# Molecular Recognition of Imidazole Derivatives by Co(III)-Porphyrins in Phosphate Buffer (pH = 7.4) and Cetylpyridinium Chloride Containing Solutions

**DOI:** 10.3390/molecules26040868

**Published:** 2021-02-06

**Authors:** Galina Mamardashvili, Elena Kaigorodova, Olga Dmitrieva, Oscar Koifman, Nugzar Mamardashvili

**Affiliations:** G.A Krestov Institute of Solution Chemistry of the Russian Academy of Sciences, Akademicheskaya st. 1, 153045 Ivanovo, Russia; gmm@isc-ras.ru (G.M.); yiii42@bk.ru (E.K.); dmitrievao.a@ya.ru (O.D.); oik@isuct.ru (O.K.)

**Keywords:** Co(III)-porphyrins, imidazole, axial coordination, selective binding, molecular recognition

## Abstract

By means of spectrophotometric titration and NMR spectroscopy, the selective binding ability of the Co(III)-5,15-bis-(3-hydroxyphenyl)-10,20-bis-(4-sulfophenyl)porphyrin (Co(III)P1) and Co(III)-5,15-bis-(2-hydroxyphenyl)-10,20-bis-(4-sulfophenyl)porphyrin (Co(III)P2) towards imidazole derivatives of various nature (imidazole (L1), metronidazole (L2), and histamine (L3)) in phosphate buffer (pH 7.4) has been studied. It was found that in the case of L2, L3 the binding of the “first” ligand molecule by porphyrinates Co(III)P1 and Co(III)P2 occurs with the formation of complexes with two binding sites (donor–acceptor bond at the center and hydrogen bond at the periphery of the macrocycle), while the “second” ligand molecule is added to the metalloporphyrin only due to the formation of the donor–acceptor bond at the macrocycle coordination center. The formation of stable complexes with two binding sites has been confirmed by density functional theory method (DFT) quantum chemical calculations and two-dimensional NMR experiments. It was shown that among the studied porphyrinates, Co(III)P2 is more selective towards to L1-L3 ligands, and localization of cobalt porphyrinates in cetylpyridinium chloride (CPC) micelles does not prevent the studied imidazole derivatives reversible binding. The obtained materials can be used to develop effective receptors for recognition, delivery, and prolonged release of drug compounds to the sites of their functioning. Considering that cetylpyridinium chloride is a widely used cationic biocide as a disinfectant, the designed materials may also prove to be effective antimicrobial agents.

## 1. Introduction

Molecular recognition processes, in which one molecule (“host” or receptor) recognizes and binds another molecule (“guest” or substrate) to form a system due to intermolecular interactions find broad application in the design of multifunctional devices for new molecular technologies. Examples are the separation of mixtures of closely related compounds and enantiomers, gas or toxic substances storing, stabilization of highly reactive compounds, the prolonged release of drugs in the conditions close to the physiological and control of reaction routes by including such particles into molecular reactors or channels [1,2,3,4,5].

Metalloporphyrin molecules are very perspective objects for molecular recognition of ions and neutral molecules. The porphyrinates are capable of forming complexes of various stability with different substrates, such as cations, anions, oxygen, nitric oxide, nitrogen and sulfur-containing organic bases, etc. [6,7,8,9,10], depending on the nature of the central metal cation. The most metalloporphyrin receptors described in the literature are hydrophobic Zn(II)-porphyrins [11,12,13,14,15,16,17], which form supramolecular complexes with substrates in the organic media (dichloromethane, chloroform, benzene, or toluene). Hydrophilic Co(III)-porphyrins are characterized by a high binding ability towards various nitrogen-containing biomolecules in aqueous, biological media [18,19,20,21].

Surfactants are widespread applications in biocatalysis, polymer science, sensors, solar cells, fuel cells, and biomass processing, and as thermal fluids and ionogels. They are used to give other compounds their tunable physiochemical and surface properties, and as by themselves therapeutic agents [22,23,24,25,26]. The purpose of this paper was to study the recognition ability of two hydrophilic Co(III)-porphyrins with hydroxyl groups in the phenyl rings of the macrocycle towards to imidazole derivatives-unsubstituted imidazole (L1), metronidazole (L2), and histamine (L3) due to possibility of formation additional hydrogen bonds between the functional groups of the receptor (porphyrin molecule) and substrate (imidazole derivative). Structures of the Co(III)-porphyrins *bis*-axial complexes with the ligands L are depicted in Figure 1.

The complexation processes in phosphate buffer (pH = 7.4) imitating blood plasma has been studied. We previously found that the more effective method of influence and control of binding and elimination processes of hydrophilic Co(III)-porphyrins with various organic bases in aqueous media is micellization, namely the localization to the outer layer (it may be intercalated among the CPC chains, most likely with the pyridinium group extending into the polar headgroup region of the micelle) surfactant micelle [21], which gives them new properties. In this regard, the investigation of the recognition ability of the studied Co(III)-porphyrins hydroxy derivatives were carried out both in phosphate buffer (pH = 7.4) and in solutions containing cetylpyridinium chloride (CPC).

## 2. Results and Its Discussion

### 2.1. Formation of the Co(III)P bis-Axial Complexes with Imidazole Derivatives

Co(III)-complexes of water-soluble tetraarylporphyrins [Co(III)P] are characterized by *bis*-axial coordination of both nitrogen and oxygen-containing small organic molecules, in which there is one additional ligand coordinated on each sides of the macrocycle plane [18,19]. Earlier, we studied the processes of Co(III)-tetra(carboxyphenyl)porphyrin (Co(III)TPPC) and Co(III)-tetra(sulfophenyl)porphyrin (Co(III)TPPS) complexation with various drugs based on nitrogen-containing heterocycles [20,21].

Co(III)P1 and Co(III)P2 form *bis*-aqua complexes in aqueous media at pH 7.4 similar to Co(III)TPPC and Co(III)TPPS. The replacement process of aqua-groups with organic bases goes sequentially in two stages:Co(III)P-(H_2_O)_2_ + L ↔ Co(III)P-(L)(H_2_O)(1a)
Co(III)P-(L)(H_2_O) + L ↔ Co(III)P-(L)_2_(1b)

Each stage of the process is accompanied by its own characteristic spectral changes in the UV–Vis spectra of the reaction system. In particular, the replacement of the “first” water molecule with the imidazole molecule is characterized by a slight decrease in the intensity of the Soret band in the UV–Vis spectrum and its red shift by 3.5 nm. The replacement of the second water molecule was accompanied by less noticeable spectral changes (Figure 2b). The calculated constants of water molecules replacement for the imidazole compounds (*K*^1^ and *K*^2^) are listed in Table 1 and Figure 3. The *bis*-axial complexes formation is confirmed by the ^1^H NMR and mass spectrometry data. The spectral changes observed with the replacement of water molecules by imidazole-containing drugs are similar to those shown in Figure 2.

The *mono*-axial complexes Co (III)P-(L) (H_2_O) are formed at a 1:1 concentration ratio of the reagents. A small excess of imidazoles is required to replace the “second” water molecule. It was found that the imidazole derivatives L2, and L3 form more stable *mono*-axial complexes with the Co (III)P(1-2), in comparison with the unsubstituted imidazole L1. This is probably due to the formation of additional binding centers between the porphyrinate and the ligand–hydrogen bonds, namely between the porphyrin hydroxyl group and the carbonyl oxygen atom of the ligand L3 or between the porphyrin hydroxyl group and the oxygen atom of the nitro group of the ligand L2.

In the reports [13,14,15,16,17], devoted to the formation of supramolecular porphyrin complexes with several binding points, the contributions of axial coordination energy (Δ*G*_axial coord._) and H-bonds (Δ*G*_HB_) to the total energy of the complex formation were estimated. A quantitative measure of increase in stability of the complex “porphyrinate-polyfunctional ligand” due to additional bond formation is *K*^1^/*K*^1,o^, where *K*^1,o^ is a stability constant of an axial metalloporphyrin complex with one binding center, and *K*^1^ is a stability constant of an axial metalloporphyrin complex containing additional binding center (or centers) (Table 2).

As seen from the Table 1, the energies of hydrogen bonds for the complexes Co(III)P1-(L2)(H_2_O), Co(III)P1-(L3)(H_2_O), Co(III)P2-(L2)(H_2_O), and Co(III)P2-(L3)(H_2_O) were in the range from 1.7 to 5.5 kJ/mol. Porphyrinate Co(III)P2-(OH)_2_ had the highest selectivity with respect to the studied imidazole derivatives upon binding of one imidazole ligand. The formation constants of Co(III)P-(L)_2_
*bis*-axial complexes according to Equation (1b) did not differ significantly from each other, i.e., no binding selectivity was observed.

The selectivity of *mono*-axial binding of histamine, metronidazole, and their precursor imidazole with Co(III)P1, Co(III)P2 is confirmed by both quantum-chemical calculations and NMR spectroscopy data. The optimization of structures by quantum-chemical calculations showed that a partial transfer of the electron pair of the Nj-atom to the empty orbitals of the Co-atom (LP N → LP × Co) is observed in the Co(III)P1, Co(III)P2 complexes with two axial ligands (L2 and L3). According to the natural bond orbital (NBO), the stabilization energy of the Co-Nj bond was 214.6–238.5 kJ/mol, and the corresponding charge transfer value was 0.215–0.259 e (Table 2).

The calculation results showed that a decrease in the Co-Nj bond length led to an increase in the binding energy E_st_ (Co-Nj) (Table 2). At the same time, an inverse relationship is observed between the length of the Co-Nj bonds and the binding energy of porphyrin with the ligand E_int_ (Table 2). The reason for this dependence is an increase in the stability of the Co(III)P-L complex not only due to Co-L donor–acceptor interactions in these complexes, but also due to the formation of additional hydrogen bonds. However, the saddle-like structure of the macrocycle led to the formation of only one additional hydrogen bond between the hydroxyl group in the phenyl ring of the porphyrinate and the oxygen atom of the carboxyl group of L2 or nitro group of L3 (Figure 4).

Thus, the strongest binding of the ligand to porphyrin (E_int_) corresponded to the smaller value of the hydrogen bond length between them (r(OH…O = L)). Therefore, the strongest bonds are formed when porphyrinate Co(III)P2 binds the metronidazole (L2). The lengths of the formed hydrogen bond and the energies of the intermolecular interaction of the *bis*-axial complex are in good agreement with the values of the energies of hydrogen bonds determined from the formation constants of the Co(III)P-(L)(H_2_O) complexes (Figure 5).

For the analysis of the chemical structure of Co(III)P(1-2) and its complexes with L1-L3, ^1^H NMR spectra were obtained as shown in the Figure 4, which are the superposition (convolution) of spectra, relating to a certain range of self-diffusion coefficients, which these bonds are characterized [17,27,28,29,30,31]. Such superposition is obtained by a projection of a pseudo-two-dimensional spectrum of the diffusive ordered DOSY spectroscopy on the set range of self-diffusion coefficients. The analysis of the obtained ranges (Figure 5) showed a change in the values of chemical shifts and a shape of the signals of characteristic groups that are usually observed as a result of the formation of complexes with ligands. In particular, formation of hydrogen bonds between one OH group of the porphyrin Co(III)P1 and oxygen atom of carboxy-group of the ligand L3 or oxygen atom of nitro-group of the ligand L2 was evidenced by the up-field shift of signals of the porphyrin OH groups, which were involved in the hydrogen bonds formation (∆ = 1.14 ppm in case of the complexes Co(III)P1(L_3_)(H_2_O), and 2.60 ppm in case of the Co(III)P1(L_2_)(H_2_O)), compared to the OH group proton of the Co(III)P1. The protons of another OH group of the Co(III)P1, which are not involved in the formation of the considered hydrogen bonds in the complexes Co(III)P1(L_3_)(H_2_O), Co(III)P1(L_2_)(H_2_O) appear in the same region (6.26 ppm) as in the initial complex Co(III)P1. Up-field shift of the Co(III)P(1-2) complexes proton signals agreed well with the DOSY experiments. The experiments were used for the compounds identification according to their self-diffusion coefficients, which depend on the size and the form of a molecule in accordance with the Stokes–Einstein equation [32,33,34,35]. The complexes Co(III)P(1-2) are characterized by a higher relative value of the self-diffusion coefficient to the solvent (0.39), than the complexes with ligands (0.34 and 0.28), which confirms the changes of molecular mass as a result of the complex formation. In addition to this, Co(III)P2 and Co(III)P2-L2, and Co(III)P2-L3 complexes were completely characterized by means of two-dimensional correlation spectroscopy (1H–1H 2D of COSY), which also is applied for the determination of macromolecules chemical structure [17,30,31,32,33,34,35].

Based on this spectroscopic technique, the signals in the one-dimensional nuclear magnetic resonance experiment were assigned to the characteristic groups of the studied compounds. As shown in Figure 6, we unambiguously assign the proton signals both Co(III)P and the corresponding ligands of Co(III)P2-L2 and Co(III)P2-L3 complexes. In addition, the nuclear Overhauser effect experiment in a rotating system of coordinates (1H–1H 2D ROESY) was carried out to confirm of the space orientation of L2 to oxygen atoms of Co(III)P2. Recently, this method has been successfully applied to define the spatial structure of macromolecules [36,37,38,39], and the small biologically active molecules [40]. This experiment unambiguously showed through space interactions of ligand protons with the Co(III)P protons located in close proximity to oxygen. The appearance of an additional cross-peak (a red circle) resulting due to the interaction of the COOH or NO_2_ groups of the ligands with the OH group of the porphyrins (Figure 6) confirms this fact.

### 2.2. Co(III)/Co(II) Red-Ox Processes in the Composition of the Co(III)P in the Presence of CPC

Axial complexes of the porphyrinates with imidazole derivatives [Co(III)P-(L)_2_] are very stable in aqueous media, and the central cation reduction Co(III)/Co(II) in the metalloporphyrin is the only method of this complexes decomposition (imidazole derivatives elimination). As shown earlier [21], such a reduction takes place in solutions containing surfactants with opposite charged porphyrin.

The study of the water-soluble porphyrin interactions with micelles of ionic surfactants revealed the presence of three different forms of porphyrins in solutions: the free porphyrins in the monomers form, the premicellar surfactant-porphyrin aggregates and the micellized monomer [21,41,42,43,44,45]. The formation of premicellar porphyrin aggregates occurs at low surfactant concentrations, and is accompanied by a decrease in the intensity and broadening of absorption bands in the UV–Vis spectra. The formation of a micellar monomer is observed with an increase in the surfactant concentration, and is accompanied by an increase in the intensity of absorption bands in the UV–Vis spectra.

Upon spectrophotometric titration of Co(III)P(1-2)-(L)_2_ porphyrin solutions with CPC (Figure 7a, Equation (2a)) with small additions of surfactants, the decrease in the absorption intensity is observed, which corresponds to the formation of an [Co(III)P(1-2)-(L)_2_]^Agg^ associate. After passing through the minimum and further increase in the CPC concentration, the intensity of the absorption bands in the corresponding UV–Vis spectra began to increase (Figure 6b, Equation (2b)), which indicates the destruction of associates and the appearance of micellar monomeric porphyrins.
Co(III)P-(L′)(L″) + CPC ↔ [Co(III)P-(L′)(L″)]^Agg^(2a)
[Co(III)P-(L′)(L″)]^Agg^ + CPC ↔ [Co(III)P-(L′)(L″)]^Mc^(2b)

As can be seen from the data of Figure 8 and Table 3, the CPC concentrations, at which the largest premicellar aggregates and micellar monomeric porphyrins (CCM) are formed, practically did not depend on the position of the hydroxyl group in the phenyl rings of Co(III)P(1-2), but significantly depended on the nature of the axial ligands. The more bulky axial ligands were coordinated on the porphyrinate, the greater the number of CPC molecules required for the formation of a complete micellar shell.

To determine the localization place of Co(III)P (preferentially in the hydrophobic regions of the micelle interior or at the water–micellar interface), and to deduce the orientation and location of the porphyrin within the micellar aggregates, the trend of the chemical shift of the surfactant (CPC) resonances in the absence and present of 0.005 M Co(III)P solution was investigated by ^1^H NMR spectroscopy. The chemical shifts of the proton resonance of CPC (0.5 M) in D_2_O/H_2_O in the presence and absence of Co(III)P are summarized in Table 4. The resonance signals of the surfactant protons occurred in the spectral region of 0.78–9.10 ppm. The chemical shifts of CPC headgroup protons (-CH- ortho-, meta-, and para-) and those of the hydrophobic alkyl chain protons (α-CH_2_-, β-CH_2_-,γ-CH_2_-, (-CH_2_-)_12_, and ω-CH_3_) indicated that the Co(III)P interacts with all groups of CPC but with various strength. The protons of the highly hydrophobic section of the molecules remain in the core portion of micelle and are highly shielded, with the NMR signals appeared in the low frequency range (in the present case protons exhibited low frequency signals at 0.95, 1.38, 1.57, and 2.12 ppm, respectively) (Figure 9a).

The CPC molecules in the micelle are mobile and can freely rotate, bend, or fold inside the micelle and are characterized by a sharply peak. However, with the addition of Co(III)P, the CPC molecule mobility inside micelles is decreased, significant broadening and changes in chemical shift are observed. This increase in line width in the presence of 50 mM Co(III)P is consistent with a reduced exchange rate at a higher concentration of CoP [46,47,48].

The protons closest to the surfactant head group and ones on the cationic head group gave rise to higher frequency NMR signals, as they were less shielded than protons located in the end of the alkyl chain. The COSY experiment of the [Co(III)P1-(H_2_O)_2_]^Mc^ system showed strong correlations between the CPC head groups protons and porphyrin phenyl rings, as evidenced by the high cross-peaks between them. Besides, there are comparatively intense cross peaks between porphyrinate protons and CPC chain protons. All these observations led us to conclude that the hydrophilic fragments of porphyrin are solubilized in the micelle head groups, and the tetrapyrrole macrocycle is located in the micelle “palisade” layer.

In the presence of surfactant micelles due to the inclusion of the porphyrin molecule in its outer micellar shell and the screening of the metalloporphyrin coordination center by the surfactant hydrocarbon chains, the Co(III)porphyrin is reduced into Co(II)porphyrin, which is accompanied by the elimination of axial ligands.

It is known that cobalt cation in the porphyrin coordination center can have two oxidation states (2^+^ and 3^+^), and both states can quite easily transform into each other. So, when Co(II)-tetraarylporphyrins are dissolved in electron-donating or non-polar solvents in the presence of organic bases, the Co(II) cation almost instantaneously oxidizes into Co(III), coordinating on itself two additional electron-donating ligands or one ligand and one solvent molecule [49,50,51,52]. The catalyst for this process is molecular oxygen contained in the air. That is, in practice, the presence of an electron donor environment and air atmosphere is the condition for finding the cobalt cation in the 3^+^ state in the corresponding porphyrinate. As noted above, when Co(III)P-(L)_2_ is localized into CPC micelle, the aqua- or imidazole complex of the porphyrinate is incorporated into the hydrophobic part of the micelle. An inert environment is created around the coordination center of the porphyrin, completely eliminating the presence of water molecules, which apparently initiates the process of Co(III) reduction into Co(II):τ, T
[Co(III)P-(L)_2_]^Mc^ ↔ [Co(II)P]^Mc^ +2L(3)

The spectral changes and reduction rate constant of micellized monomer complexes [Co(III)P-(H_2_O)_2_]^Mc^, [Co(III)P-(L)(H_2_O)]^Mc^ are presented in Table 5 and Figure 10.

Our investigations have shown that the Co(III)/Co(II) reduction rate in the micellized porphyrinate depends on various factors such as the surfactant nature, buffer ionic strength, temperature, and axial ligand nature. Co-porphyrinates with aqua-ligands have the highest reduction rate. The reduction rate of Co(III)/Co(II) is decreased at the presence of one unsubstituted imidazole ligand by 1.5 times. The two-sites binding of imidazole derivative depending on strength of additional hydrogen bond makes the Co(III)-porphyrins more stable to reduction from 2 to 4 times. *Bis*-axial complexes of [Co(III)P-(L)_2_]^Mc^ are reduced especially slow.

### 2.3. The Effect of Micelle Formation on the Processes on Axial Coordination of the Organic Ligands by the Co(III)-Porphyrins

Similarly to the monomeric *bis*-aqua complex Co(III)P-(H_2_O)_2_, the micellized [Co(III)P-(H_2_O)_2_]^Mc^ and [Co(II)P(1-2)]^Mc^ complexes in aqueous media at pH 7.4 replaces the water ligands in two stages and this process is accompanied by characteristic spectral changes (Figure 11a). The “second” ligand attachment leads to oxidation of Co(II) into Co(III) in the case of [Co(II)P(1-2)] (Figure 11b).
[Co(III)P-(H_2_O)_2_]^Mc^ + L ↔ [Co(III)P-(L)(H_2_O)]^Mc^(4a)
[Co(III)P-(L)(H_2_O)]^Mc^ + L ↔ [Co(III)P-(L)_2_]^Mc^(4b)
[Co(II)P]^Mc^ + L ↔ [Co(III)P-(L)(H_2_O)]^Mc^(4c)

The formation constants of *mono*-axial complexes of [Co(III)P(1-2)(L)(H_2_O)]^Mc^ obtained according with the equilibrium 4a confirm that selective ability of imidazole derivative binding by Co-porphyrin in CPC micellar solutions was retained (Figure 12). As the Co-porphyrin monomer, as micellized porphyrin binding ligands L2 and L3 via formation of donor–acceptor and hydrogen bonds.

## 3. Materials and Methods

*5,15-bis-(methoxyphenyl)-10,20-bis-phenylporphyrins* (**1**,**2**) were obtained according to [53] through acid catalyzed condensation reaction between meso-phenyldipyrrolylmethane and corresponding methoxy-benzaldehydes followed by oxidation of reaction mixture by *o*-chloranil. 5,15-bis-(3-methoxyphenyl)-10,20-bis-phenylporphyrin (**1**). To a solution of 0.2 g (0.96 mmol) meso-phenyldipyrrolylmethane in 50 mL of dichloromethane, were added 0.14 g (1.21 mmol) of 3-methoxybenzaldehyde in 25 mL dichloromethane and 1 mL of trichloroacetic acid. The solution was refluxed for 30 min and subjected oxidation by treatment with 0.1 g of *o*-chloranil. The reaction mixture was washed with water, evaporated to a minimum volume and chromatographed on aluminum oxide using dichloromethane as an eluent. Yield of the compound (**1**) was equal to 52%. R_f_ = 0.42 (Silufol, eluent—dichloromethane). UV–Vis spectrum in dichloromethane, λ, nm (lgε): 419.0 (4.92), 515.0 (3.60), 550.0 (3.12), 593.0 (2.87), 649.0 (2.21). ^1^H NMR in CD_2_Cl_2_: 9.28 (d, 8*H*, β-pyrrole), 7.70 (m, 8*H*, *o*-H), 7.51 (m, 10H, p-, m-H), 3.18 (s, 6*H*, OCH_3_), −2.14 (br.s., 2*H*, NH). 

*5,15-bis-(2-methoxyphenyl)-10,20-bis-phenylporphyrin* (**2**) was obtained similarly (**1**). Yield of the compound (**2**) was equal to 39%. R_f_ = 0.39 (Silufol, eluent—dichloromethane). UV–Vis spectrum in dichloromethane, λ, nm (lgε): 417.0 (4.89), 513.0 (3.58), 548.0 (3.10), 591.0 (2.83), 647.0 (2.19). ^1^H NMR in CD_2_Cl_2_: 9.22 (d, 8*H*, β-pyrrole), 7.66 (m, 6*H*, *o*-H), 7.48 (m, 12*H*, p-, m-H), 3.14 (s, 6*H*, OCH_3_), −2.09 (br.s., 2*H*, NH).

*5,15-bis-(hydroxyphenyl)-10,20-bis-phenylporphyrins* (**3**,**4**) were obtained according with [54] by demethylation of the 5,15-bis-(methoxyphenyl)-10,20-bis-phenylporphyrins (**1**,**2**) under action of trimethylsilane iodide in chloroform in the presence of hydrochloric acid according. 5,15-bis-(3-hydroxyphenyl)-10,20-bis-phenylporphyrin (**3**). To a solution of 0.3 g (0.39 mmol) of porphyrin (**1**) in 50 mL of chloroform with stirring in argon atmosphere was added 0.1 g (0.46 mmol) of trimethylsilane iodide in 20 mL chloroform. To the solution was added 15 mL of 5% hydrochloric acid and the resulting solution was heated at 70 °C for 30 min. The reaction mixture was cooled, washed with water, evaporated to a minimum volume, and chromatographed on aluminum oxide using dichloromethane as an eluent. Yield of the compound (**3**) was equal to 62%. Rf = 0.22 (Silufol, eluent—mixture of dichloromethane:acetone, 3:1. UV–Vis spectrum in dichloromethane, λ, nm (lgε): 412.0 (4.71), 509.0 (3.43), 541.0 (3.01), 587.0 (2.79), 641.0 (2.11). ^1^H NMR in D_2_O/H_2_O: 9.12 (d, 8*H*, β-pyrrole), 7.62 (m, 8*H*, *o*-H), 7.45 (m, 8*H*, p-, m-H), 2.01 (br.s., 2*H*, OH). 5,15-bis-(2-hydroxyphenyl)-10,20-bis-phenylporphyrin (**4**) was obtained similarly (**3**). Yield of the compound (**8**) was equal to 49%. R_f_ = 0.1 (Silufol, eluent—mixture of dichloromethane:acetone, 3:1). UV–Vis spectrum in dichloromethane, λ, nm (lgε): 410.0 (4.69), 507.0 (3.41), 539.0 (2.98), 584.0 (2.72), 639.0 (2.09). ^1^HNMR in D_2_O/H_2_O: 9.10 (d, 8*H*, β-pyrrole), 7.62 (m, 6*H*, *o*-H), 7.45 (m, 10*H*, p-, m-H), 2.01 (br.s., 2*H*, OH).

*5,15-bis-(hydroxyphenyl)-10,20-bis-(sulfophenyl)porphyrins* (**5**,**6**) were prepared as described in [55] from the intermediate products (3, 4) respectively. 5,15-bis-(3-hydroxyphenyl)-10,20-bis-(4-sulfophenyl)porphyrin (**5**). Of 5,15-bis-(3-hydroxyphenyl)-10,15-bis-phenylporphyrin 0.15 g (0.19 mmol) was dissolved by stirring in 30 mL of concentrated sulfuric acid at room temperature and heated at 70 °C for 30 min. The resulting solution was cooled poured onto ice, the precipitate was filtered and washed with water and used without further purification. Yield of the compound (**5**) was equal to 87%. 5,15-bis-(2-hydroxyphenyl)-10,15-bis-(4-sulfophenyl)porphyrin (**6**) was obtained similarly to (**5**). Yield of the compound (**6**) was equal to 79%. The compounds 5 and 6 were used in subsequent transformations without additional purification.

Structures of the porphyrins (**1**–**6**) are depicted in Figure 13.
Co(III)P-(L)(H_2_O) + L ↔ Co(III)P-(L)_2_ (1b)(5)

*Co(II)-5,15-bis-(hydroxyphenyl)-10,20-bis-(4-sulfophenyl)porphyrins [Co(II)P1, Co(II)P2]* were obtained according to the literature procedure [56] from the corresponding porphyrin ligands (**3**,**4**) and Co(II) acetate by refluxing in DMF. Co(II)-5,15-bis-(3-hydroxyphenyl)-10,20-bis-(4-sulfophenyl)porphyrin (Co(II)P1)**.** To 0.1 g (mmol) solution of compound **5** in 50 mL of dimethylformamide was added 100 mg of cobalt(II) acetate and resulting mixture was refluxed for 30 min. The resulting solution was diluted with water, the precipitate formed was filtered, dried, and chromatographed on aluminum oxide using toluene as an eluent. Yield of the Co(II)P1 was equal to 79%. Rf = 0.62 (aluminum oxide, eluent—toluene). UV–Vis spectrum in DMF, λ, nm (lgε): 414.2 (5.37), 528.3 (4.24). ^1^H NMR in DMSO-*d*_6_: 15.87 (br. s., 8*H*, β-pyrrole), 12.96 (br.s., 4*H*, 4-sulfophenyl), 10.08 (br. s., 4*H*, 4-sulfophenyl), 11.82 br.s., 4*H*, *o*-H), 9.65 (m, br.s., 4*H*, p-, m-H), 8.27 (s, -OH). Mass spectrum, *m*/*z* (I_rel_, %): 865.1 (94) [M]^+^. Calculated for C_44_H_26_N_4_S_2_O_8_Co 865.9. 

*Co(II)-5,15-bis-(2-hydroxyphenyl)-10,20-bis-(4-sulfophenyl)porphyrin [Co(II)P2]* was obtained similarly to Co(II)P1. ^1^H NMR in DMSO-*d*_6_: 16.03 (br.s., 8*H*, β-pyrrole), 13.05 (br.s., 4*H*, 4-sulfophenyl), 10.14 (br.s., 4*H*, 4- sulfophenyl), 11.67 (br.s., 2*H*, *J* = 8.1 Hz *o*’-H), 11.60 (t, 2*H*, *m*’-H), 9.29 (t, 2*H*, *p*-H), 9.10 (d, 2*H*, *J* = 8.1 Hz, *m*-H), 5.61 (s, -OH). UV–Vis spectrum in DMF, λ, nm (lgε): 413.2 (5.35), 527.3 (4.23). Mass spectrum, *m*/*z* (I_rel_, %): 864.8 (92) [M]^+^. Calculated for C_44_H_26_N_4_S_2_O_8_Co: 865.9.

Co(III)-porphyrins [Co(III)-5,15-bis-(3-hydroxyphenyl)-10,20-bis-(4-sulfophenyl)porphyrin (Co(III)P1-(H_2_O)_2_) and Co(III)-5,15-bis-(2-hydroxyphenyl)-10,20-bis-(4-sulfophenyl)porphyrin (Co(III)P2-(H_2_O)_2_)] were obtained similarly [56] by dissolving of corresponding Co(II)-porphyrins ([Co(II)P1, Co(II)P2] in aqueous media.

Co(III)-5,15-bis-(3-hydroxyphenyl)-10,20-bis-(4-sulfophenyl)porphyrin (Co(III)P1-(H_2_O)_2_. 1H NMR in D_2_O/H_2_O: 9.07 (s, 8*H*, β-pyrrole), 8.26 (d, 4*H*, *J* = 8.2 Hz, 4-sulfophenyl), 8.16 (d, 4*H*, *J* = 8.1 Hz, 4-sulfophenyl), 7.81 (m, 4*H*, *o*-H), 7.65 (m, 4*H*, *p*-, *m*-H), 6.27 (s, 2*H*, HO-Ar). UV–Vis spectrum in phosph. buf., λ, nm (lgε): 424.0 (5.33), 540.0(4.09).

*Co(III)P1-(L2)(H_2_O)*: ^1^H NMR in D_2_O/H_2_O: 9.05 (s, 8*H*, β-pyrrole), 8.22 (d, 4*H*, *J* = 8.2 Hz, 4-sulfophenyl), 8.11 (d, 4*H*, *J* = 8.1 Hz, 4-sulfophenyl), 7.79 (m, 4*H*, *o*-H), 7.63 (m, 4*H*, *p*-, *m*-H), 6.25 (s, 1H, HO-Ar); 5.05 (br.s, 1*H*, -CH_2_-CH_2_OH), 4.37 (t, 2*H*, -CH_2_-CH_2_OH), 3.79 (t, 2*H*, -CH_2_-CH_2_OH), 3.67 (s, 1*H*, HO-Ar); 3.25 (s, 1*H*, 2-Im), 2.28 (s, 3*H*, -CH_3_). UV–Vis in phosph. buf., λ,nm (lgε): 426.0 (5.31); 543.0 (4.06).

*Co(III)P1-(L3)(H_2_O)*: ^1^H NMR in D_2_O/H_2_O: 9.04 (s, 8*H*, β-pyrrole), 8.21 (d, 4*H*, *J* = 8.2 Hz, 4-sulfophenyl), 8.10 (d, 4*H*, *J* = 8.1 Hz, 4-sulfophenyl), 7.80 (m, 4*H*, *o*-H), 7.62 (m, 4*H*, *p*-, *m*-H), 6.26 (s, 1*H*, HO-Ar), 6.14 (br.s. 2*H* (NH_2_), 6.03 (s, H, Im), 5.22 (br.s, H, -COOH), 5.12 (s, 1*H*, HO-Ar), 4.68 (s, *H*, -CH<), 4.18 (m, 2*H*, -CH_2_-), 3.65 (s, *H*, Im). UV–Vis in phosph. buf., λ, nm (lgε): 425.5 (5.29), 542.0 (4.05).

*Co(III)-5,15-bis-(2-hydroxyphenyl)-10,20-bis-(4-sulfophenyl)porphyrin (Co(III)P2-(H_2_O)_2_)*. ^1^H NMR in D_2_O/H_2_O: 9.05 (s, 8*H*, β-pyrrole), 8.25 (d, 4 *H*, *J*=8.2 Hz, 4-sulfophenyl), 8.14 (d, 4*H*, *J* = 8.1 Hz, 4-sulfophenyl), 7.67 (d, 2*H*, *J* = 8.1 Hz *o*’-H), 7.60 (t, 2H, *m*’-H), 7.19 (t, 2*H*, *p*-H), 7.10 (d, 2*H*, *J* = 8.1 Hz, *m*-H), 5.61 (s, -OH). UV–Vis in phosph. buf., λ, nm (lgε): 423.0 (5.29); 539.0 (3.97).

*Co(III)-5,15-bis-(2-hydroxyphenyl)-10,20-bis-(4-sulfophenyl)porphyrin (Co(III)P2-(H_2_O)_2_)*. ^1^H NMR in D_2_O/H_2_O: 9.05 (s, 8*H*, β-pyrrole), 8.25 (d, 4 *H*, *J* = 8.2 Hz, 4-sulfophenyl), 8.14 (d, 4*H*, *J* = 8.1 Hz, 4-sulfophenyl), 7.67 (d, 2*H*, *J* = 8.1 Hz *o*’-H), 7.60 (t, 2*H*, *m*’-H), 7.19 (t, 2H, *p*-H), 7.10 (d, 2*H*, *J* = 8.1 Hz, *m*-H), 5.61 (s, -OH). UV–Vis in phosph. buf., λ, nm (lgε): 423.0 (5.29); 539.0 (3.97).

*Co(III)P1-(L2)_2_*. ^1^H NMR in D_2_O/H_2_O: 9.07 (s, 8*H*, β-pyrrole), 8.25 (d, 4*H*, *J* = 8.2 Hz, 4-sulfophenyl), 8.14 (d, 4*H*, *J* = 8.1 Hz, 4-sulfophenyl), 7.80 (m, 4*H*, *o*-H), 7.64 (m, 4*H*, *p*-, *m*-H), 6.20 (s, *H*, -OH); 5.02 (br.s, 2*H*, -CH_2_-CH_2_OH), 4.33 (m, 4*H*, -CH_2_-CH_2_OH), 3.64 (m, 4*H*, -CH_2_-CH_2_OH), 3.23 (s, 2*H* 2-Im), 2.27 (s, 6*H*, -CH_3_). UV–Vis in phosph. buf., λ, nm (lgε): 427.6 (5.30); 547.0 (4.04).

*Co(III)P1-(L3)_2_*. ^1^H NMR in D_2_O/H_2_O: 9.07 (s, 8*H*, β-pyrrole), 8.26 (d, 4*H*, *J* = 8.2 Hz, 4-sulfophenyl), 8.16 (d, 4*H*, *J* = 8.1 Hz, 4-sulfophenyl), 7.81 (m, 4*H*, *o*-H), 7.65 (m, 4*H*, *p*-, *m*-H), 6.20 (s, *H*, -OH), 6.14 (br.s. 4*H* (NH_2_), 6.03 (s, *H*, Im), 5.22 (br.s, 2*H*, -COOH), 5.12 (s, *H*, -OH), 4.68 (s, 2*H*, -CH<), 4.18 (m, 4*H*, -CH_2_-), 3.65, 3.60 (s, 2*H*, Im). UV–Vis in phosph. buf., λ, nm (lgε): 427.0 (5.27); 546.0 (4.02).

*Co(III)P2-(L2)(H_2_O)*. ^1^HNMR in D_2_O/H_2_O: 9.05 (s, 8*H*, β-pyrrole), 8.25 (d, 4*H*, *J* = 8.2 Hz, 4-sulfophenyl), 8.14 (d, 4*H*, *J* = 8.1 Hz, 4-sulfophenyl), 7.67 (d, 2*H*, *J* = 8.1 Hz *o*’-H), 7.60 (t, 2*H*, *m*’-H), 7.19 (t, 2*H*, *p*-H), 7.10 (d, 2*H*, *J* = 8.1 Hz, *m*-H), 5.61 (s, *H*, -OH), 5.06 (br.s, 1*H*, -CH_2_-CH_2_OH), 4.37 (t, 2*H*, -CH_2_-CH_2_OH), 3.80 (t, 2*H*, -CH_2_-CH_2_OH), 3,.67 (s, 1*H*, -OH), 3.25 (s, 1*H*, 2-Im), 2.28 (s, 3*H*, -CH_3_). UV–Vis in phosph. buf., λ, nm (lgε): 425.0 (5.28); 542.0 (3.97).

*Co(III)P2-(L2)_2_*. ^1^HNMR in D_2_O/H_2_O: 9.05 (s, 8*H*, β-pyrrole), 8.25 (d, 4*H*, *J* = 8.2 Hz, 4-sulfophenyl), 8.14 (d, 4*H*, *J* = 8.1 Hz, 4-sulfophenyl), 7.67 (d, 2*H*, *J* = 8.1 Hz *o*’-H), 7.60 (t, 2*H*, *m*’-H), 7.19 (t, 2*H*, *p*-H), 7.10 (d, 2*H*, *J* = 8.1 Hz, *m*-H), 5.61 (s, *H*, -OH), 5.03 (br.s, 2*H*, -CH_2_-CH_2_OH, 4.33 (m, 4*H*, -CH_2_-CH_2_OH), 3.63 (m, 4*H*, -CH_2_-CH_2_OH), 3.20 (s, 2*H* 2-Im), 2.26 (s, 6*H*, -CH_3_). UV–Vis in phosph. buf., λ, nm (lgε): 426.5 (5.27); 545.6 (3.94).

*Co(III)P2-(L3)(H_2_O)*. ^1^HNMR in D_2_O/H_2_O: 9.05 (s, 8*H*, β-pyrrole), 8.25 (d, 4*H*, *J* = 8.2 Hz, 4-sulfophenyl), 8.14 (d, 4*H*, *J* = 8.1 Hz, 4-sulfophenyl), 7.67 (d, 2*H*, *J* = 8.1 Hz *o*’-H), 7.60 (t, 2*H*, *m*’-H), 7.19 (t, 2*H*, p-H), 7.10 (d, 2*H*, *J* = 8.1 Hz, m-H), 6.32 s(*H*, -OH), 6.13 (br.s. 4*H* (NH_2_), 6.04 (s, *H*, Im), 5.26 (br.s, *H*, -COOH), 5.15 s(*H*, -OH); 4.67 (s, *H*, -CH<), 4.16 (m, 2*H*, -CH_2_-), 3.63 (s, *H*, Im). UV–Vis in phosph. buf., λ, nm (lgε): 424.0 (5.25); 541.0 (3.96).

*Co(III)P2-(L3)_2_*. ^1^HNMR in D_2_O/H_2_O: 9.05 (s, 8*H*, β-pyrrole), 8.25 (d, 4*H*, *J* = 8.2 Hz, 4-sulfophenyl), 8.14 (d, 4 *H*, *J* = 8.1 Hz, 4-sulfophenyl), 7.67 (d, 2*H*, *J* = 8.1 Hz *o*’-H), 7.60 (t, 2*H*, m-H), 7.19 (t, 2*H*, p-H), 7.10 (d, 2*H*, *J* = 8.1 Hz, m-H), 6.20 (s, *H*, -OH), 6.14 (br.s. 4*H* (NH_2_), 6.03 (s, *H*, Im, 5.22 (br.s, 2*H*, -COOH), 5.12 (s, *H*, -OH), 4.68 (s, 2*H*, -CH<), 4.18 (m, 4*H*, -CH_2_-), 3.65, 3.60 (s, 2*H*, Im). UV–Vis in phosph. buf., λ, nm (lgε): 426.0 (5.23); 544.5 (3.95).

Structures of the complexes Co(III)P(1-2)(L2)(H_2_O), Co(III)P(1-2)(L3)(H_2_O), Co(III)P(1-2)(L2)_2_, and Co(III)P(1-2)(L3)_2_ are depicted in Figure 14.

### 3.1. Spectrophotometric Studies

Thermodynamic constants for the complexation of the Co(III)P with L were calculated according to the Equation (6) on the base of the spectrophotometric titration experiment results:(6)$$K=\frac{\left[CoP-L\right]}{\left[CoP\right]\cdot \left[L\right]}=\frac{1}{\left[L\right]}\cdot \left(\frac{\Delta A_{i,\;\lambda_{1}}}{\Delta A_{0,\;\lambda_{1}}}\cdot \frac{\Delta A_{0,\;\lambda_{2}}}{\Delta A_{i,\;\lambda_{2}}}\right)\left(M^{-1}\right)$$
where *λ*_1_ is descending wavelength, *λ*_2_ is ascending wavelength; [Co(III)P-Im] is the concentration of the porphyrinate with one axial ligand; and [*L*] is the ligand concentration. Δ*A*_0_ is the maximal change of solution optical density at the given wavelength, and Δ*Ai* is the change of solution optical density at the given wavelength at the given concentration [57].

Kinetic parameters of the investigated reaction were obtained according to the known procedure [33]. Effective rate constants (*k*_eff_) were determined by the change in the solution optical density on working wave lengths (*λ* = 414, 425 nm) after definite time intervals by the equation of formally first order (Equation (7)) when CPC were in excess.
*k*_eff_ = 1/τ log(*c*_0_/*c*_τ_)(7)
where *c*_0_ and *c*_τ_ are the complexes concentrations of at start of the process and at time τ. Values *k*_eff_ and determination of average deviations were optimized with using of Microsoft Excel and ggh.exe (QB-45) by the Guggenheim method. Relative error was 3–5%.

### 3.2. NMR Studies

NMR experiments were performed on a Bruker Avance III 500 MHz NMR spectrometer (Bruker Biospin, Karlsruhe, Baden-Württemberg, Germany) equipped with a 5-mm probe using standard Bruker TOPSPIN Software. Temperature control was performed using a Bruker variable temperature unit (BVT-2000) in combination with a Bruker cooling unit (BCU-05) to provide chilled air. Experiments were performed at 298 K without sample spinning. TMS signals were used as internal standards for counting chemical shifts. The two-dimension diffusion ordered spectroscopy (2D DOSY) spectra were recorded with PGSTE pulse sequence using a bipolar gradient pulses and the insertion of a supplementary delay (LED) [58]. The PGSTE sequence was used with a diffusion delay of 0.1 s, a total diffusion-encoding pulse width of 5 ms. For each of 32 gradient amplitudes, 32 transients of 16.384 complex data points were acquired. The two-dimensional correlation spectroscopy (2D COSY) spectra with a zero-quantum suppression element [59] were acquired with a 16.96 ppm spectral window in the direct dimension F1 with 2048 complex data points and a 16.96 ppm spectral window in the indirect dimension F2 with 128 complex points. The spectra were acquired with 64 scans and relaxation delay of 2 s. Two-dimensional rotating frame nuclear Overhauser effect spectroscopy (2D ROESY) [59] experiments were performed with pulsed filtered gradient techniques. The spectra were recorded in a phase-sensitive mode using Echo/Antiecho-TPPI gradient selection with 2048 points in the F2 direction and 256 points in the F1 direction. Spin-lock delay values for 2D ROESY were 200 ms. The spectra were acquired with 64–72 scans and relaxation delay of 2 s.

### 3.3. Quantum-Chemical Calculations

Were performed using the Gaussian 09 software package [60]. The optimization of the geometrical parameters of the compounds under study and NBO-analysis were performed using the density functional theory method (DFT) with the introduction of dispersion interactions in the semiempirical Grimm DFT-D3 (BJ) model [61]. We used the CAM-B3LYP functional [62] with the basic set: Def2TZVPP [63,64] for the Co atom and 6-31G (d, p) [65,66] for the rest of the atoms. All structures did not have virtual frequencies in vibration spectra correspond to energy minimum.

The energy of intermolecular interaction of the studied complexes ΔE with taking into account the superposition error of BSSE was calculated according to the Equation (8) [67,68]:ΔE = *E*(*AB*,*aUb*,*R*) − [*E*(*A*,*aUb*,*R*) + *E*(*B*,*aUb*,*R*)](8)

The error arising from the superposition of basis sets of functions (BSSE) was calculated using the Equation (9):BSSE = [*E*(*A*,*aUb*,*R*) − E(A,a,R)] + [E(*B*,*aUb*,*R*) − *E*(*B*,*b*,*R*)](9)
where *E*(*AB*,*aUb*,*R*),*E*(*A*,*a*,*R*), and *E*(*B*,*b*,*R*) are the energy of the complex and the initial molecules, respectively. Molecules *A* and *B* are separated by the distance *R* in the complex, *a* and *b* are the basic set of isolated molecules, and *aUb* is the basic set of the *AB* complex.

The stabilization energy of the electronic pair acceptor orbitals at the bond formation ($E_{st}$) and the values of charge transfer ($q_{CT}$) were calculated by the following Equation (10) [69,70]:(10)$$E_{st}=\frac{-2F_{ij}^{2}}{\Delta E}$$
$$q_{CT}=2{\left(\frac{F_{ij}}{\Delta E}\right)}^{2}$$
where $F_{ij}$—the Fock matrix element between *i* and *j* NBO orbitals and ΔE—the difference in orbital energies.

## 4. Conclusions

Thus, it was found that disulfoderivatives of Co(III)-tetraarylporphyrines with hydroxyl groups in two aryl fragments of the macrocycle have the ability to recognize the various imidazole derivatives (including drugs) due to the formation of additional hydrogen bonds upon their axial coordination on the cobalt cation in aqueous media. The strongest additional hydrogen bonds were formed in the case of Co(III)-porphyrinate with a hydroxyl group in the ortho-position of the phenyl fragment and the nitro group of the pharmaceutical metronidazole molecule. Moreover, the selectivity of binding was recorded when only one imidazole derivative was attached in axial-position. It was found that the localization of the investigated sulfonated Co(III)-tetraarylporphyrin into spherical micelles of cetylpyridine chloride decreased their binding ability towards imidazoles, but the selectivity of their binding was retained.

Micellation of the Co(III)-porphyrins leads to a complete or partial reduction of the central cobalt cation into Co(II). The rate of the reduction process and the degree of its occurrence very strongly depended on the strength of the axial ligands binding: bis-aqua Co(III)-porphyrins were reduced more rapidly, and *bis*-imidazole complexes were reduced more slowly. The presence of additional hydrogen bonds during the formation of *mono*-axial complexes of Co(III)-porphyrins with imidazole derivatives increased their resistance to reduction by 2–4 times. The rate of reduction can also be controlled by increasing the temperature, varying the nature of substrate or medium. The obtained materials can be used to develop effective receptors for recognition, delivery, and prolonged release of drug compounds to the sites of their functioning. Considering that cetylpyridinium chloride is a widely used cationic biocide as a disinfectant, the materials we developed might prove to be effective antimicrobial agents. We are currently conducting research in this area.

## Figures and Tables

**Figure 1 molecules-26-00868-f001:**
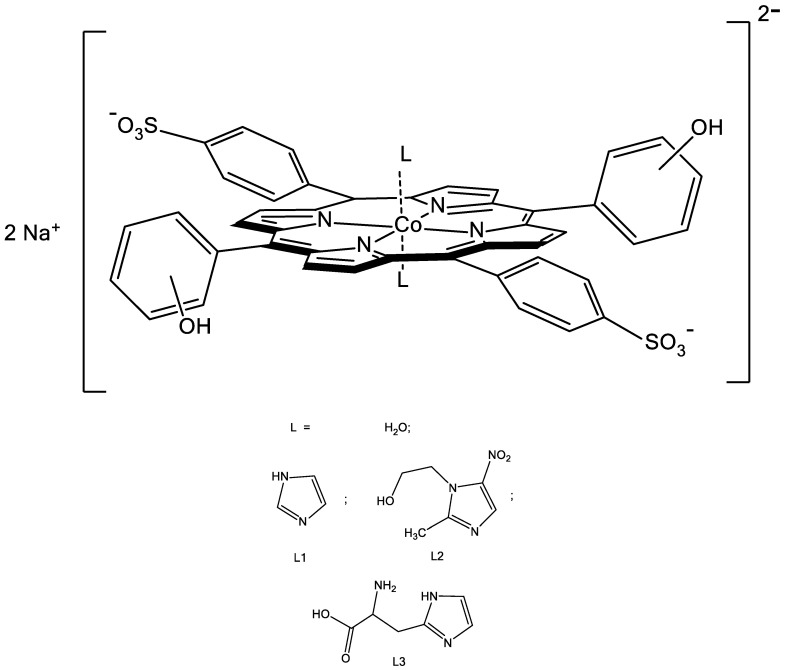
Structures of the Co(III)-porphyrins *bis*-axial complexes with the ligands (L = H_2_O, L1-L3).

**Figure 2 molecules-26-00868-f002:**
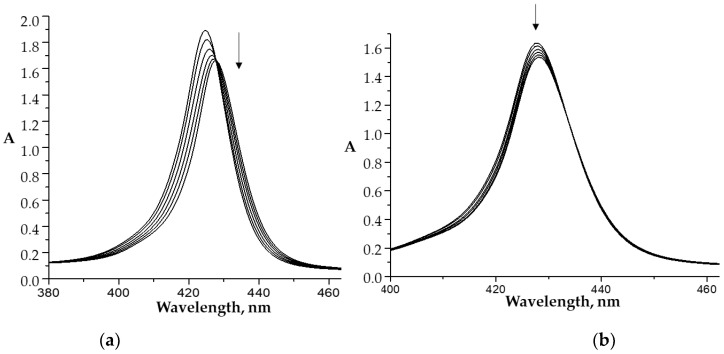
UV–Vis spectra (Soret band region) of the CoP1(H_2_O)_2_ (C_porph._ − 9.8 × 10^−6^ M) in phosphate buffer (pH 7.4) at 25 °C upon titration with L1 (C_L1_ = 0 ÷ 5.9 × 10^−6^ M (**a**) and C_L1_ = 5.9 × 10^−6^ ÷ 1.9 × 10^−4^ M (**b**)).

**Figure 3 molecules-26-00868-f003:**
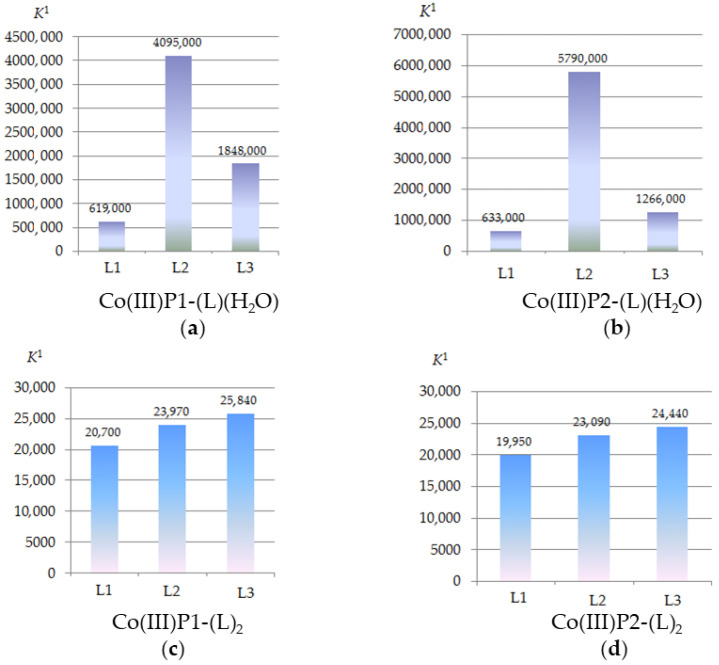
Stability constants (*K*^1^ and *K*^2^, (M^–1^) of the *mono-* and *bis*-axial complexes Co(III)P(1-2)-(L)(H_2_O) and Co(III)P(1-2)-(L)_2_, phosphate buffer (pH 7.4) at 25 °C.

**Figure 4 molecules-26-00868-f004:**
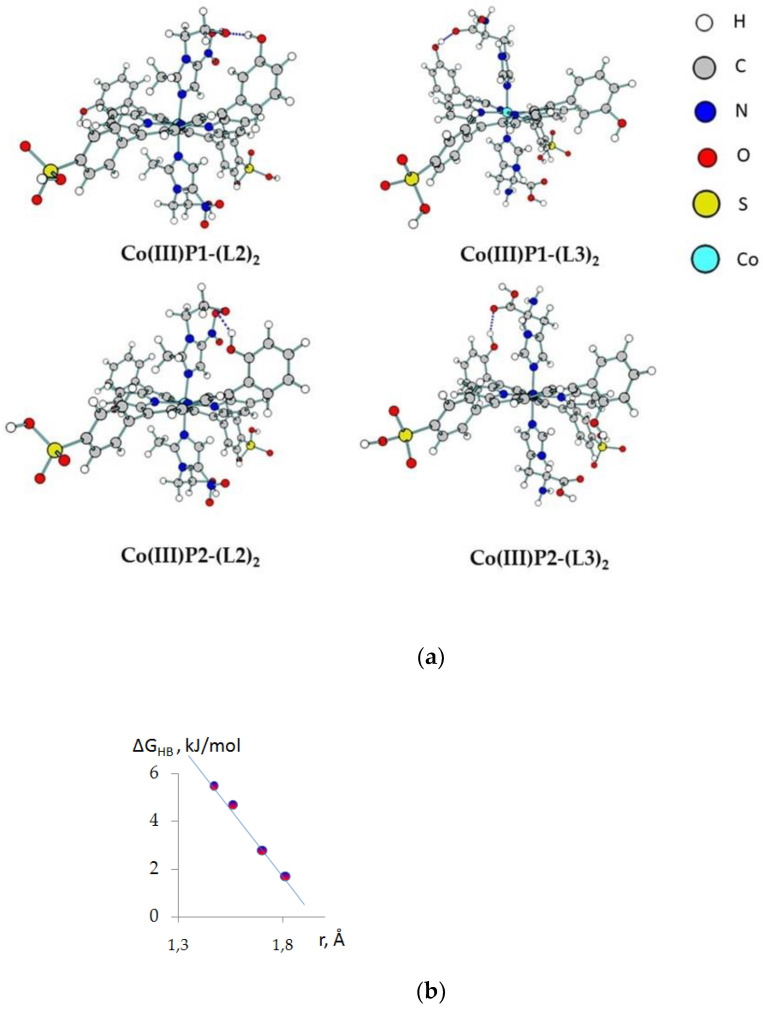
Optimized structures of *bis*-axial complexes (**a**) and dependence of the length of the additional hydrogen bond in these complexes with ΔG_HB_, calculated from experimental data (**b**).

**Figure 5 molecules-26-00868-f005:**
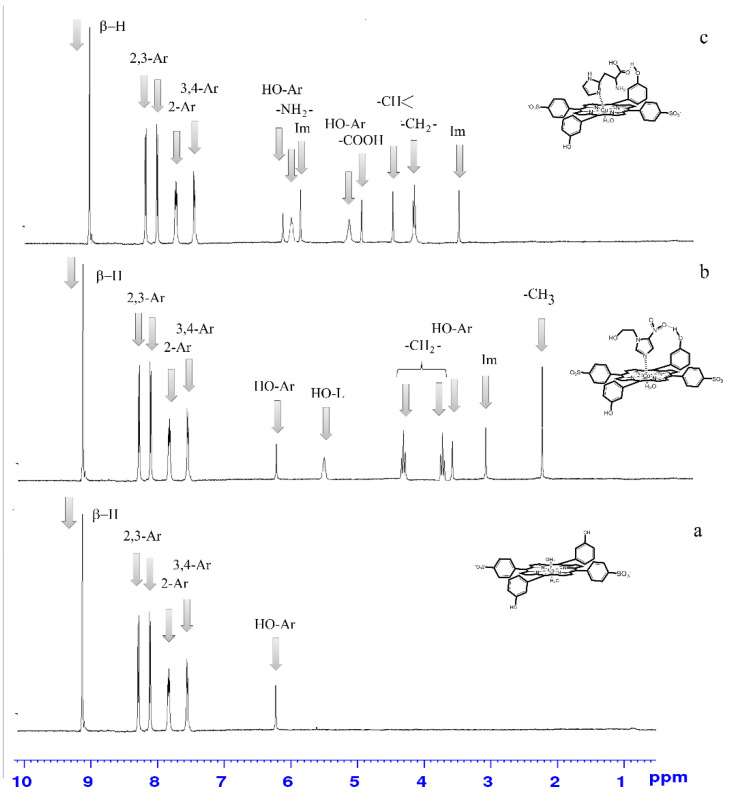
1D-(part) of the 2D DOSY NMR spectra of the porphyrinate Co(III)P1(H_2_O)_2_ (**a**), Co(III)P1(L_2_)(H_2_O) (**b**), and Co(III)P1(L_3_)(H_2_O) (**c**) in D_2_O/H_2_O, corresponding to the characteristic self-diffusion coefficient of these complexes.

**Figure 6 molecules-26-00868-f006:**
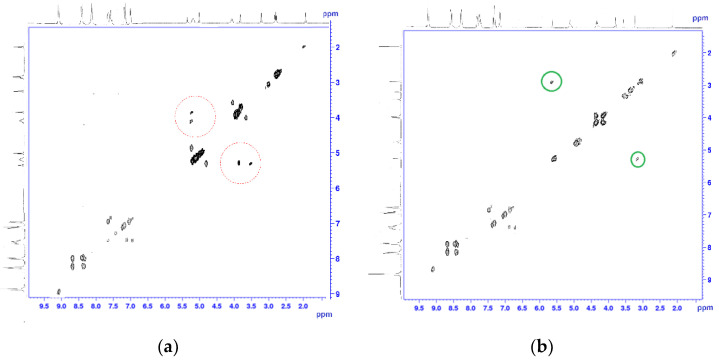
^1^H–^1^H 2D ROSY NMR spectra of the complexes Co(III)P2-(L2) (**a**) and Co(III)P2-(L3) (**b**) in D_2_O/H_2_O.

**Figure 7 molecules-26-00868-f007:**
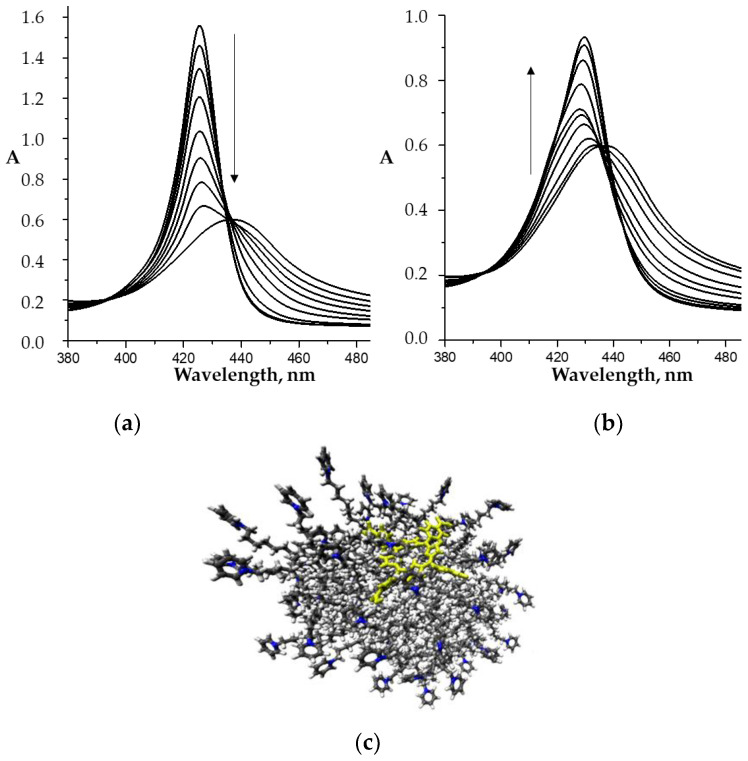
UV–Vis spectra (Soret band region) of the Co(III)P(1-2)-(L1)_2_ in phosphate buffer (pH 7.4) 25 °C upon titration with CPC: (**a**) −C_CPC_ = 0 ÷ 2 × 10^–5^ M; (**b**) −C_CPC_ = 2 × 10^-5^ ÷ 4 × 10^–4^ M); and (**c**) schematic image of the micelle structure [Co(III)P(1-2)-(L′)(L″)]^CPC^.

**Figure 8 molecules-26-00868-f008:**
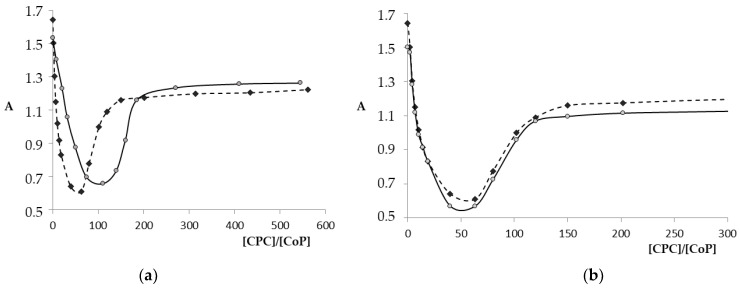
Co(III)P-(L′)(L″) titration curves for CPC in phosphate buffer (pH 7.4) at the decreasing (426 nm) and increasing (436 nm) wavelengths: (**a**) Co(III)P1-(L2)(H_2_O) (ο) and Co(III)P1-(L2)_2_ (♦) and (**b**) Co(III)P1-(L2)(H_2_O) (ο) and Co(III)P2-(L2)(H_2_O) (♦).

**Figure 9 molecules-26-00868-f009:**
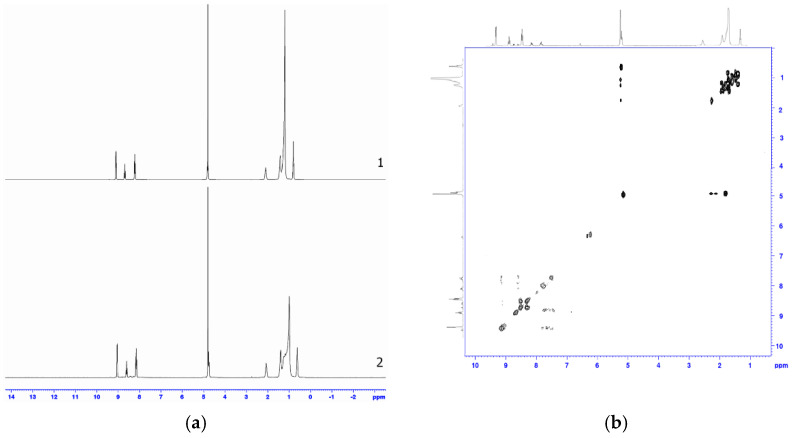
(**a**) ^1^H-NMR-spectrum of the CPC in the absence (1) and in the presence of Co(III)P1 (2) (D_2_O/H_2_O) in the region of 0.5–2.5 ppm and (**b**) ^1^H–^1^H two-dimensional correlation spectroscopy (2D COSY) NMR spectra of the [Co(III)P1-(H_2_O)_2_]^Mc^, C_CPC_ = 6 × 10^−3^ M, C_Co(III)P_ = 1 × 10^−4^ M.

**Figure 10 molecules-26-00868-f010:**
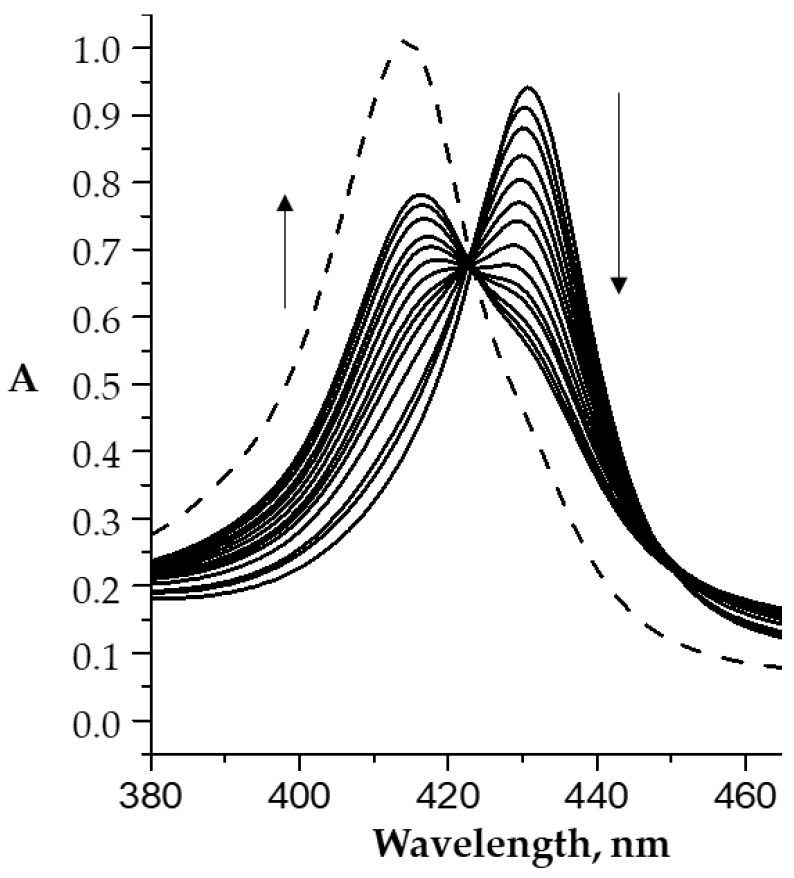
The reduction processes of [Co(III)P1(H_2_O)_2_]^Mc^ into [Co(II)P1^Mc^ in phosphate buffer (pH 7.4) during 150 min at T = 40 °C.

**Figure 11 molecules-26-00868-f011:**
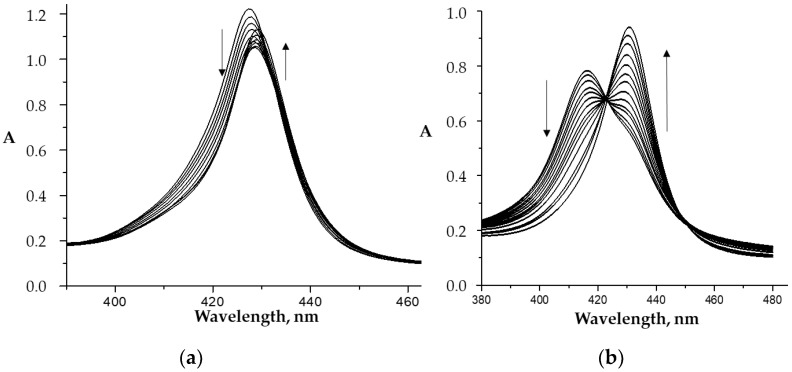
UV–Vis spectra (Soret band region) of the [Co(III)P1-(H_2_O)_2_]^Mc^ (**a**) and [Co(II)P1]^Mc^ (**b**) upon titration with L1 (C_CPC_ = 0 ÷ 8 × 10^–4^ M, in phosphate buffer (pH 7.4) at 25 °C).

**Figure 12 molecules-26-00868-f012:**
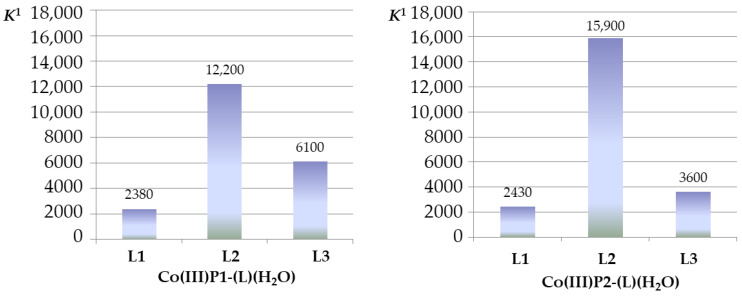
Stability constants (*K*^1^ and *K*^2^, (M^−1^) of the micellized *mono*-axial porphyrinates Co(III)P1-(L)(H_2_O)]^Mc^ and [Co(III)P2-(L)(H_2_O)]^Mc^ in phosphate buffer (pH 7.4) at 25 °C.

**Figure 13 molecules-26-00868-f013:**
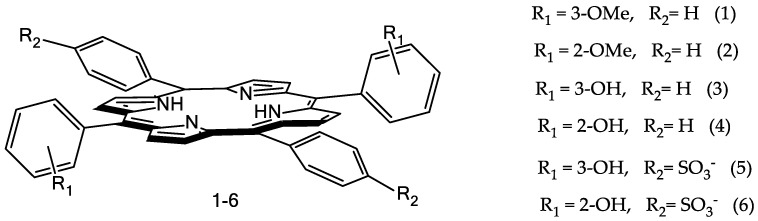
Structures of the porphyrins (**1–6**).

**Figure 14 molecules-26-00868-f014:**
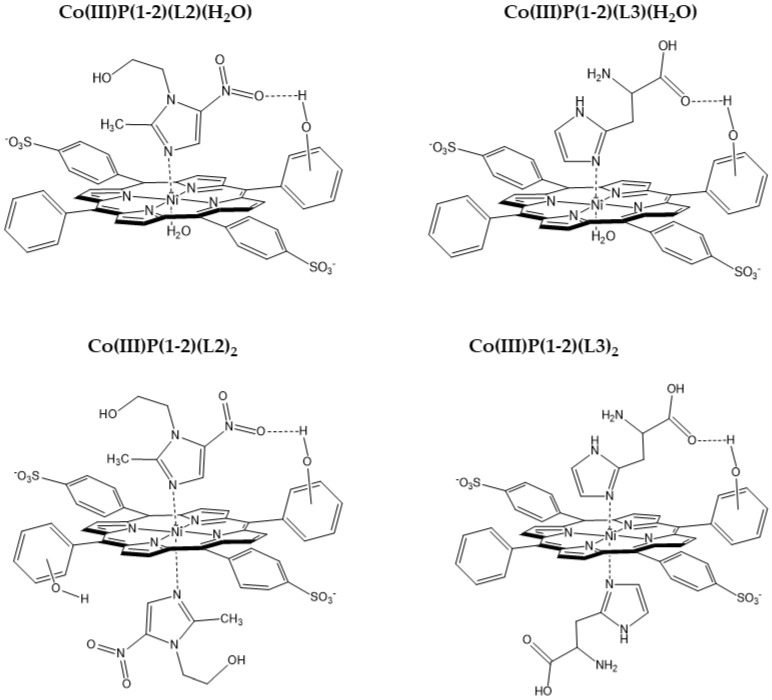
Structures of the complexes Co(III)P(1-2)(L2)(H_2_O), Co(III)P(1-2)(L3)(H_2_O), Co(III)P(1-2)(L2)_2_, and Co(III)P(1-2)(L3)_2_.

**Table 1 molecules-26-00868-t001:** Energies of Co(III)P-(L)(H_2_O) and Co(III)P-(L)(H_2_O) formation (−ΔG1 and −ΔG2, kJ × mol^−1^), calculated values of the Gibbs energy of the *bis*-axial complexes (−ΔG2, kJ × mol^−1^), length (r, Å), and energy (−ΔG_HB_, kJ × mol^−1^) of hydrogen bonds.

Co(III)P and L	−ΔG^1 a^	−ΔG^2 b^	−ΔG_HB_ ^c^	−ΔG^Co(III)P-(L)^_2_ ^d^	r, Å (OH⋅⋅⋅O-L)
Co(III)P1/L1	33.0	24.6	-	216.2	-
Co(III)P1/L2	37.7	25.0	4.7	295.4	1.56
Co(III)P1/L3	35.8	25.2	2.8	222.6	1.70
Co(III)P2/L1	33.1	24.5	-	215.4	-
Co(III)P2/L2	38.6	24.9	5.5	309.7	1.47
Co(III)P2/L3	34.8	25.0	1.7	216.9	1.81

^a^ The total energy of Co(III)P-(L)(H_2_O) complexes formation according to Equation (1a) (ΔG = −RT ln*K*^1^). ^b^ The total energy of Co(III)P-(L)_2_ complexes formation from Co(III)P-(L)_2_ according to Equation (1b) (ΔG = −RT ln*K*^2^). ^c^ The energy of hydrogen bonding in the complex Co(III)P-(L)(H_2_O) (ΔG_HB_ = ΔG ^Co(III)P-L^ − ΔG ^Co(III)P-L1^) (L-ligand (L1), the compound that does not form hydrogen bonds). ^d^ The calculated value of the hydrogen bond length (OH⋅⋅⋅O-L; r, Å).

**Table 2 molecules-26-00868-t002:** Geometrical and energetic parameters of Co(III)P-L.

System	Co-N_p_, Ǻ	Co-N_j_, Ǻ	*r* (OH…O = L), °	*E*_st_(Zn-Nj) kJ/mol	*q*_CT_, *e*	−*E*_int_(BSSE) kJ/mol
Co(III)P1-(L2)_2_	1.924	1.937	1.56	−216.4	0.242	295.4
Co(III)P1-(L3)_2_	1.934	1.929	1.80	−238.5	0.218	222.6
Co(III)P2-(L2)_2_	1.929	1.930	1.47	−214.6	0.259	309.7
Co(III)P2-(L3)_2_	1.946	1.922	1.81	−242.7	0.215	216.9

N_p_—nitrogen atom of Co(III)P. N_j_—nitrogen atom of imidazole ligand (L_2_ and L_3_).

**Table 3 molecules-26-00868-t003:** CPC concentrations at which premicellar aggregates and micellated porphyrinates are formed (C_porph._ = 1.15 × 10^−5^ M).

	C_CPC_^Agg^_,_M	cmc *, M (N^a^ = C_CPC_/C_porph._)
Co(III)P1-(H_2_O)_2_	4.61 × 10^−5^	9.43 × 10^−4^ (82)
Co(III)P2-(H_2_O)_2_	4.67 × 10^−5^	9.55 × 10^−4^ (83)
Co(III)P1-(L1)(H_2_O)	4.77 × 10^−4^	1.52 × 10^−3^ (132)
Co(III)P1-(L2)(H_2_O)	5.22 × 10^−4^	1.70 × 10^−3^ (148)
Co(III)P1-(L3)(H_2_O)	5.84 × 10^−4^	1.77 × 10^−3^ (154)
Co(III)P1-(L1)_2_	1.12 × 10^−3^	2.18 × 10^−3^ (190)
Co(III)P1-(L2)_2_	1.28 × 10^−3^	2.47 × 10^−3^ (215)
Co(III)P1-(L3)_2_	1.32 × 10^−3^	2.58 × 10^−3^ (224)

* cmc—the critical concentration of the localized porphyrin molecule of the CPC micelle.

**Table 4 molecules-26-00868-t004:** Chemical shift of CPC protons in D_2_O/H_2_O in the presence and absence of Co(III)P1 (C_CPC_ = 1 × 10^−4^ M).

[Co(III)P]	m-	p-	o-	α-CH_2_-	β-CH_2_-	γ-CH_2_-	(-CH_2_-)_13_		ω-CH_3_
0	9.08	8.67	8.19	4.75	2.12	1.57	1.38		0.95
0.001	9.10	8.69	8.22	4.74	2.10	1.41	1.20		0.78
Δδ	0.02	0.02	0.03	−0.01	−0.02	−0.16	−0.18		−0.17

**Table 5 molecules-26-00868-t005:** Reduction rate constants of [Co(III)P-(H_2_O)_2_]^Mc^, [Co(III)P-(L)(H_2_O)]^Mc^, and [Co(III)P-(L)_2_]^Mc^ into [Co(II)P]^Mc^ in phosphate buffer (pH 7.4) (*k_eff_*, s^−1^·M^−1^, C_CPC_ = 2.35 × 10^−3^ M, T = 40 °C).

	[Co(III)P-(H_2_O)_2_]^Mc^	[Co(III)P-(L)(H_2_O)]^Mc^	[Co(III)P-(L)_2_]^Mc^
Co(III)P1/L1	2.75 × 10^−4^	1.84 × 10^−4^	<10^−5^
Co(III)P1/L2		5.77 × 10^−5^	
Co(III)P1/L3		7.92 × 10^−5^	
Co(III)P2/L1	2.71 × 10^−4^	1.80 × 10^−4^	
Co(III)P2/L2		4.07 × 10^−5^	
Co(III)P2/L3		7.85 × 10^−5^	

## Data Availability

Not applicable.

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
