# Peer review of "Molecular Recognition of Imidazole Derivatives by Co(III)-Porphyrins in Phosphate Buffer (pH = 7.4) and Cetylpyridinium Chloride Containing Solutions"

_molecules, 2021, doi:10.3390/molecules26040868_

Round 1

Reviewer 1 Report

The article «Molecular recognition of imidazole derivatives by Co(III)-porphyrins in phosphate buffer (pH=7.4) and cetylpyridinium chloride containing solutions» by authors Galina Mamardashvili, Elena Kaigorodova, Olga Dmitrieva, Oscar Koifman, and Nugzar Mamardashvili is devoted to studying of the recognition ability of hydrophilic Co(III)-porphyrins towards to imidazole, metronidazole, and histamine in phosphate buffer (pH=7.4), imitating blood plasma. Taking into the account that micellization is an effective method of influence and control of binding and elimination processes of metalloporphyrins with various organic bases, in this article the authors conducted the investigation of the Co(III)-porphyrins hydroxyl-derivatives recognition of N-containing guest molecules in solutions containing cetylpyridinium chloride (CPC). It was found that disulfide derivatives of Co(III)-tetraarylporphyrines with hydroxyl groups in two aryl fragments of the macrocycle have the ability to recognize the studied imidazole derivatives (including drugs) due to the formation of additional hydrogen bonds upon their axial coordination on the cobalt cation in aqueous media. The strongest additional hydrogen bonds are formed in the case of Co(III)-porphyrinate with a hydroxyl group in the ortho-position of the phenyl fragment and the nitro group of the pharmaceutical metronidazole molecule. Moreover, the selectivity of binding was recorded when only one imidazole derivative was attached in axial-position. It was found that the localization of the investigated sulfonated Co(III)-tetraarylporphyrin into spherical micelles of cetylpyridine chloride decreases their binding ability towards imidazoles, but the selectivity of their binding is retained.

The strength of the study is that the authors established that the micellation of the Co(III)-porphyrins leads to a reduction of the central cobalt cation into Co(II). The rate of the reduction process and the degree of its occurrence very strongly depends on the strength of the axial ligands binding: bis-aqua Co(III)-porphyrins are reduced most rapidly, and bis-imidazole complexes are reduced most slowly. It is shown for the first time that the presence of additional hydrogen bonds during the formation of mono-axial complexes of Co(III)-porphyrins with imidazole derivatives increases their resistance to reduction by 2-4 times. The rate of reduction can also be controlled by increasing the temperature, varying the nature of substrate or medium.

As a side note, it should be noted that:

1) It is not clear from the text of the article how complete the transformation of the ligand porphyrin into cobalt-porphyrinate occurs. Are there traces of unreacted initial porphyrin ligand in the products of the complexation reaction?

2) Missing references 16,17 in the introduction on the page 2 (second paragraph)

3) According to the requirements of the journal Molecules, it is necessary to give DOI for all references

In my point of view, the results of the study make a significant contribution to the chemistry of substrate-receptor interactions and may be of interest to researchers working in the field of design of effective receptors for recognition and prolonged release of bioactive compounds to the sites of their functioning. Considering that CPC is a widely used cationic biocide as a disinfectant, the materials may prove to be effective antimicrobial agents.

The article should be published after correcting the indicated comments.

Reviewer 2 Report

The manuscript entitled “Molecular recognition of imidazole derivatives by Co(III)-porphyrins in phosphate buffer (pH=7.4) and cetylpyridinium chloride containing solutions” describes binding features of Co(III)-porphyrins decorated with hydroxyphenyl and sulfonatophenyl meso-groups towards bio-relevant imidazole derivatives. Guest binding stoichiometry, topologies and selectivity are demonstrated using various spectroscopic studies and theoretical calculations.  Moreover, the Co(III)-porphyrins are described to form spherical micelles in presence of cetylpyridine chloride surfactant.

This manuscript is a collection of large volume of data without precise presentation and rationalization. The manuscript is distractive and found many flaws. Thus, I could not recommend acceptance of this manuscript in current form. The manuscript should be more concise. Some critical points can be noted below:

  1. The author should discuss about the synthetic methods at the beginning (results and discussions), especially if the porphyrins and respective Co(III) complexes are new.
  2. A synthetic scheme should be provided describing the chemical structures, reagents etc. in the main manuscript. Yield for precursor porphyrins 5 and 9 are mentioned as 85% in both cases which is surprisingly high under the reported reaction condition. Please recheck the yield. Also check the 1H NMR of 5 and 9 in CDCl3/CD2Cl2 to monitor the core NH protons.
  3. Figure 3: Optimized structures are misleading, Two sets of meso-aryl groups looks like carboxylate rather than sulfonate.
  4. Complexation of imidazole ligand and porphyrins are monitored by 1H NMR spectroscopy. However, exact changes of chemical shift values upon ligand complexation are not mentioned. As from Figure 4, changes in Ar-OH protons which are involved in hydrogen bonding seems minimal.  

Few typical and other errors are mentioned below among many:

 In conclusion section: “…disulfide derivatives of Co(III)-tetraarylporphyrines”: which one is disulfide derivative?

Second sentence in the abstract should be modified. A lengthy statement is included in the parentheses. 

Co(II)P2 1H NMR in DMSO-d6 rather than DMSO

Line 86 pyridine molecule or imidazole? Inconsistent with Figure 1

Line 87 Red shifted   

Figure 1 caption “at 25oC”

Figure 9, 10 caption: Wavelength (nm)

Round 2

Reviewer 1 Report

The authors completely corrected all mistakes and responded to the comments. I suppose that the manuscript can be accepted for publication

Reviewer 2 Report

Authors tried to modify the sections as questioned by this reviewer. However some portions of this revised manuscript remain error-ridden. Please revise the manuscript thoroughly before being published. Few points are noted below:

  1. The graphical abstract does not reflect a clear summary or objective of this work. 
  2. In the synthetic details: “…condensation of corresponding meso-aryldipyrrolylmethanes with meso-phenyldipyrrolylmethane..” this statement is wrong. The whole sentence is also grammatically incorrect. Revise.
  3. Line 431: “Yield of the compound (5) was equal to 52%” compound (1) rather than 5. Line 436: compound 2 rather than 9.
  4. The author modified the DFT optimized structures in Fig. 3. Do the corresponding structural parameters change? explain.

Author Response

Thank you very much for carefully reading of our article and making very valuable comments.
